DOI: 10.1038/s41467-018-04351-0　　**OPEN**

# Transient metal-centered states mediate isomerization of a photochromic ruthenium-sulfoxide complex

Amy A. Cordones [1], Jae Hyuk Lee[2], Kiryong Hong [3], Hana Cho[4], Komal Garg[5], Martial Boggio-Pasqua [6], Jeffrey J. Rack[5,7], Nils Huse [8,9], Robert W. Schoenlein[1,10] & Tae Kyu Kim [11]

Ultrafast isomerization reactions underpin many processes in (bio)chemical systems and molecular materials. Understanding the coupled evolution of atomic and molecular structure during isomerization is paramount for control and rational design in molecular science. Here we report transient X-ray absorption studies of the photo-induced linkage isomerization of a Ru-based photochromic molecule. X-ray spectra reveal the spin and valence charge of the Ru atom and provide experimental evidence that metal-centered excited states mediate isomerization. Complementary X-ray spectra of the functional ligand S atoms probe the nuclear structural rearrangements, highlighting the formation of two metal-centered states with different metal-ligand bonding. These results address an essential open question regarding the relative roles of transient charge-transfer and metal-centered states in mediating photoisomerization. Global temporal and spectral data analysis combined with time-dependent density functional theory reveals a complex mechanism for photoisomerization with atomic details of the transient molecular and electronic structure not accessible by other means.

[1] PULSE Institute, SLAC National Accelerator Laboratory, Stanford University, Menlo Park, CA 94025, USA. [2] Pohang Accelerator Laboratory, Pohang 37673, Republic of Korea. [3] Center for Gas Analysis, Division of Chemical and Medical Metrology, Korea Research Institute of Standards and Science, Daejeon 34113, Republic of Korea. [4] Center for Analytical Chemistry, Division of Chemical and Medical Metrology, Korea Research Institute of Standards and Science, Daejeon 34113, Republic of Korea. [5] Nanoscale and Quantum Phenomena Institute, Department of Chemistry and Biochemistry, Ohio University, Athens, OH 45701, USA. [6] Laboratoire de Chimie et Physique Quantiques, CNRS et Université de Toulouse, 118 route de Narbonne, UMR 5626, IRSAMC, Toulouse 31062, France. [7] Department of Chemistry & Chemical Biology, The University of New Mexico, Albuquerque, NM 87131, USA. [8] Institute for Nanostructure and Solid State Physics, Department of Physics, University of Hamburg, Hamburg 22761, Germany. [9] Max Planck Institute for the Structure and Dynamics of Matter and Center for Free-Electron Laser Science, Hamburg 22761, Germany. [10] LCLS, SLAC National Accelerator Laboratory, Menlo Park, CA 94025, USA. [11] Department of Chemistry and Chemistry Institute of Functional Materials, Pusan National University, Busan 46241, Republic of Korea. These authors contributed equally: Amy A. Cordones, Jae Hyuk Lee, Kiryong Hong, Hana Cho. Correspondence and requests for materials should be addressed to N.H. (email: nils.huse@uni-hamburg.de) or to R.W.S. (email: rwschoen@slac.stanford.edu) or to T.K.K. (email: tkkim@pusan.ac.kr)

Ultrafast photo-isomerization reactions are central to chemical processes ranging from the first step in vision[1,2], to molecular switches and logic[3–5], to photoactive materials[6–8], and photo-thermal energy conversion[9–11]. Efficient molecular isomerization involves correlated changes in the electronic and atomic structure, but we presently lack a detailed mechanistic understanding of how these correlated structural dynamics influence reaction rates and quantum efficiencies[12,13]. New insight concerning the related electronic and nuclear motion during isomerization is essential for optimization and control of such photochemical phenomena through directed synthesis.

As a particular class of materials to which isomerization is often highly relevant, photochromic transition metal complexes have demonstrated photo-responsive electronic, magnetic, and structural properties[14–18]. They provide an important framework to investigate the coupling of electronic and nuclear motion that underpins linkage isomerizations and more generally, reactions of molecular photoswitches, molecular machines, and photocatalysts. Moreover, the manipulation of their intrinsic properties and dynamics via associated ligand structure and choice of metal center can be greatly aided by mechanistic insight at the atomic level. As an example, Ruthenium-based polypyridine sulfoxide complexes exhibit Ru–S → Ru–O isomerization (Fig. 1a) that can be optically triggered with high quantum efficiency[17,19–22]. Their isomerization rates are empirically tunable via ligand modification of the functional moieties, which in this study comprise a sulfoxide group attached to a chelate ring[23].

Ultrafast optical spectroscopy studies of several sulfoxide-containing Ruthenium complexes[20,21,23–25] first revealed the dynamics of the isomerization process, and established that the initial excited state is predominantly metal-to-ligand charge-transfer (MLCT) in character. Moreover, the transient optical data of these complexes suggested photoisomerization mechanisms mediated by triplet MLCT ($^3$MLCT) excited states. For some sulfoxide complexes (e.g., those with monodentate dimethyl sulfoxide ligands), this S-bonded state ($^3$MLCT$_S$) can directly convert to the O-bonded charge transfer state ($^3$MLCT$_O$)[26]. That is, the isomerization occurs along the excited state $^3$MLCT potential energy surface (PES), as depicted in Fig. 1b[27]. Accordingly, the linkage photo-isomerization can be described as an adiabatic process, as the Ru–S bond breaking and Ru–O bond making processes occur on the lowest electronically excited PES.

For other sulfoxide complexes (e.g., those with chelating sulfoxide ligands similar to that shown in Fig. 1a), the $^3$MLCT$_S$ state was suggested to couple non-adiabatically to the singlet ground state[20,21,23–25]. This was suggested as an efficient pathway to formation of the O-bonded ground state ($^1$G$_O$), as shown in Fig. 1c, possibly via a sideways-bonded $^3$MLCT intermediate ($^3$MLCT$_{SO}$). In this scenario, the Ru–S bond breaking and Ru–O bond making processes involve both the lowest triplet and singlet PESs. However, the transient optical spectroscopy that is the basis for the isomerization mechanisms described above provides only indirect information on transient valence charge and atomic structure and is blind to transitions involving optically dark states, including metal-centered excited states that may be of particular interest for this class of systems, as described below.

An essential open question in functional transition–metal complexes is the relative roles of transient charge-transfer states (characterized by valence charge in the ligand $\pi^*$ orbitals) vs. metal-centered (MC) excited states (with valence charge in the

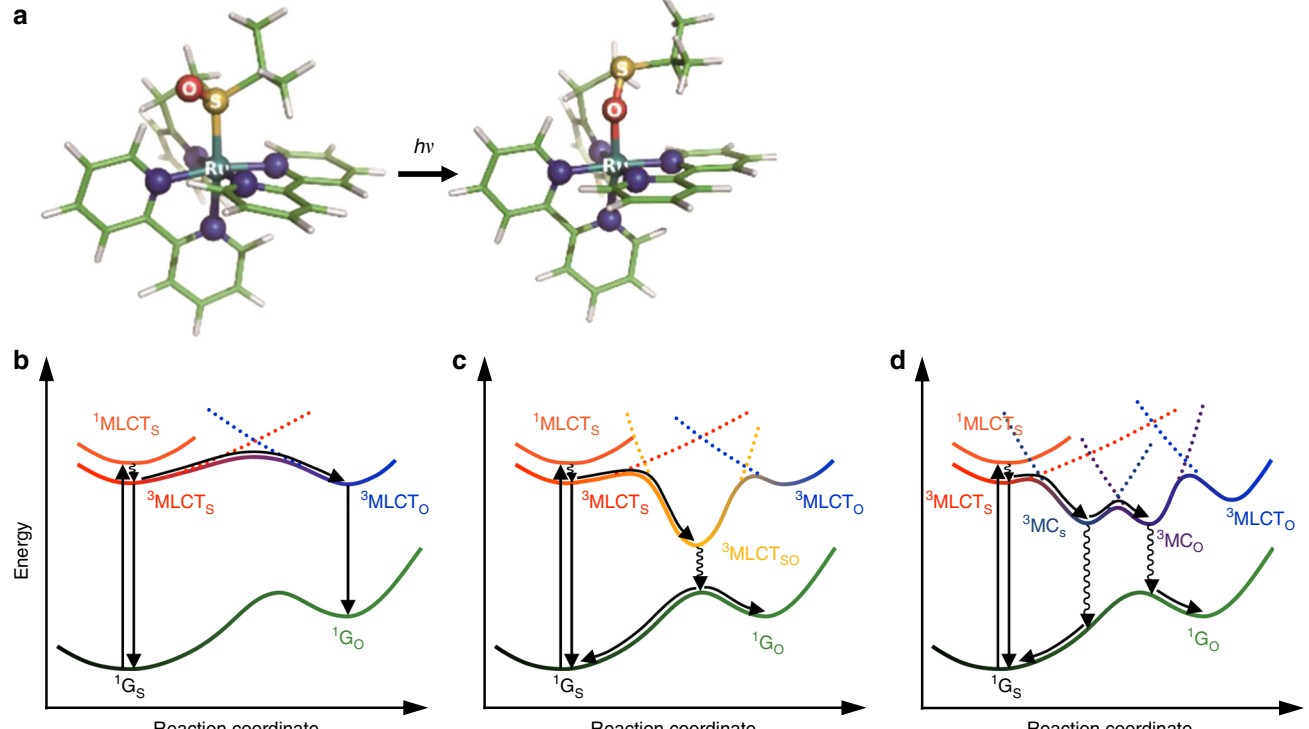

**Fig. 1** Structures of [Ru(bpy)$_2$(pyESO)]$^{2+}$ and various photoisomerization pathways. **a** Molecular structures before and after photoinduced isomerization. Color code: cyan Ru, yellow S, red O, blue N, green C, white H. **b** Adiabatic photo-isomerization on lowest $^3$MLCT excited states. **c** Non-adiabatic photoisomerization on lowest $^3$MLCT excited states. **d** Non-adiabatic photo-isomerization involving $^3$MLCT and $^3$MC excited states. Vertical arrows represent photo-absorption and emission processes. Wavy arrows represent non-radiative transitions (conical intersections are not shown for simplicity). Curvy arrows represent adiabatic relaxation paths on potential energy surfaces. Dotted lines represent diabatic states for triplet states. $^1$G$_S$: S-bonded ground state reactant, $^1$G$_O$: O-bonded ground state photoproduct, $^3$MLCT$_S$: S-bonded $^3$MLCT excited state, $^3$MLCT$_O$: O-bonded $^3$MLCT excited state, $^3$MLCT$_{SO}$: sideways SO-bonded $^3$MLCT excited state, $^3$MC$_S$: S-bonded $^3$MC excited state, $^3$MC$_O$: O-bonded $^3$MC excited state

ligand-field-split metal $d$ orbitals) in mediating the quantum efficiency of photo-induced processes such as isomerization. Theoretical studies of the Ru sulfoxide (and phosphinidene oxide) photochromic complexes, based on unrestricted density functional theory (DFT), conclude that triplet MC ($^3$MC) states mediate the isomerization (as depicted in Fig. 1d), and that degenerate potential energy crossing points occur around these MC states[22,28,29]. Thus these predictions support a model in which non-adiabatic isomerization pathways through conical intersection seams involving $^3$MC states are favored over adiabatic routes, with important implications for the speed and quantum efficiency of these reactions. Other non-adiabatic isomerization pathways that are instead mediated by $^3$MLCT states have been suggested for Ru sulfoxide complexes, based on optical spectroscopic studies[20,21,23–25], and for a Ru nitrosyl complex, based on theoretical studies[30–33]. The precise role of MC states and the PESs on which isomerization occurs, therefore, remain important open questions for this class of systems.

Time-resolved X-ray absorption spectroscopy (TR-XAS) is a powerful tool for addressing these issues since X-rays probe transient valence states from specific atomic core levels of well-defined energy and symmetry, and are sensitive to both the optically allowed CT states and optically dark MC states. In particular, transition metal L-edge spectroscopy probes unoccupied molecular orbitals with mainly metal $d$ character, while ligand K-edge spectroscopy probes molecular orbitals of predominately ligand atom $p$ character (Fig. 2), thus providing complementary views of the valence charge dynamics from the perspective of the central metal atom and from the ligand cage. Moreover, TR-XAS spectra can be directly compared with DFT and time-dependent DFT (TD-DFT) calculations to provide new details on transient valence charge distributions and transition-state molecular structures[34–39]. This approach is particularly effective in $4d$ transition-metal complexes (e.g., Ru-based) where strong spin-orbit coupling dominates and multiplet effects are negligible, rendering TD-DFT calculations a reliable tool for quantitative predictions of X-ray spectra[36].

Here we present TR-XAS studies of a photochromic Ru–sulfoxide complex, [Ru(bpy)$_2$(pyESO)]$^{2+}$ (Fig. 1a, where bpy = 2,2′-bipyridine and pyESO=2-((isopropylsulfinyl)ethyl)pyridine), using X-ray energies resonant with both the central Ru atom and S atom of the functional ligand. Transient Ru L$_3$-edge spectra provide detailed insight to the electronic structure of the ground state and excited state intermediates during isomerization. Simultaneously, S K-edge spectra probe the structural changes as the metal bonding partner switches from sulfur to oxygen. We suggest a detailed mechanism for the complex

isomerization reaction based on global analysis and comparison of our TR-XAS results with TD-DFT simulations. Importantly, our experimental results provide the first direct evidence that isomerization in [Ru(bpy)$_2$(pyESO)]$^{2+}$ proceeds from MC ligand-field states. In these states, electronic occupation of antibonding e$_g$* orbitals promotes elongation of the metal-ligand bond and facilitates the isomerization reaction.

## Results

**General.** Static and transient differential X-ray absorption spectra were measured at the Ru L$_3$-edge (2.84 keV) and at the S K-edge (2.47 keV) of [Ru(bpy)$_2$(pyESO)]$^{2+}$. Differential spectra were measured as a function of time delay following optical excitation of the $^1$MLCT transition, with time-resolution determined by the 70 ps X-ray pulse duration. Static absorption spectra representing the S-bonded species in its ground electronic state are shown in the top panels of Fig. 3a, b. The Ru L-edge spectrum is comprised of transitions from $2p$ core levels to unoccupied orbitals of primarily Ru $4d$ character ($dx^2$–$y^2$ and $dz^2$), as illustrated in Supplementary Fig. 1. The S K-edge spectrum is comprised of transitions from the S $1s$ core level to antibonding orbitals of primarily sulfoxide ligand character (with some metal character contributing to the lower energy peak), as illustrated in Supplementary Fig. 2. The differential absorption spectra measured at 100 ps delay are presented immediately below the static spectra in Fig. 3a (Ru L$_3$-edge) and Fig. 3b (S K-edge). Figure 3c, d present the time dynamics of the dominant spectral feature at the Ru L$_3$-edge (2838.5 eV) and S K-edge (2474.8 eV), respectively.

Also shown in Fig. 3a, b are the TD-DFT simulation results. The predicted transitions for the S-bonded ground state ($^1$G$_S$) are overlaid with the ground state XAS data in the top panels of Fig. 3a, b, highlighting the validity of the TD-DFT simulation approach. The predicted differential XAS spectra for several possible excited state intermediates and the O-bonded ground state (as suggested by previous optical and DFT studies) are shown in the lower panels of Fig. 3a, b: S-bonded MLCT state ($^3$MLCT$_S$), S-bonded MC state ($^3$MC$_S$), O-bonded MC state ($^3$MC$_O$), and the O-bonded ground state ($^1$G$_O$) photoproduct. The predicted differential spectra shown in Fig. 3 are scaled to fit the $^1$G$_O$ predicted differential spectrum to that measured after completion of the photoisomerization reaction (at time delay = 2.5 ns). Note that from the perspective of the Ru site (Fig. 3a) MLCT and MC states can be clearly differentiated, but the metal-ligand bonding configuration of the two MC states is indistinguishable. Conversely, from the perspective of the S site (Fig. 3b), the TD-DFT predicted differential spectra for the $^3$MC$_S$ and $^3$MC$_O$ states are clearly distinguishable in amplitude, as they represent very different sulfur coordination states, i.e., bound vs. unbound to the metal center.

**Ru L$_3$-edge TR-XAS.** Absorption at the Ru L$_3$-edge preferentially probes dipole-allowed transitions to the $4d$ levels (split into three occupied t$_{2g}$ and two unoccupied e$_g$ orbitals for octahedral Ru$^{II}$ complexes), and is thus sensitive to changes in occupancy of the Ru valence orbitals. It is clear from visual comparison (Fig. 3a) that the differential spectrum measured at 100 ps time delay does not match the simulated MLCT state spectrum ($^3$MLCT$_S$). MLCT excitation is characterized by the transfer of electron density from the Ru $4d$ (t$_{2g}$) orbitals to a bipyridyl ligand π* orbital. The predicted $^3$MLCT$_S$ transient spectrum presented here is consistent with the measured and calculated $^3$MLCT spectra of other polypyridyl Ru complexes reported previously[35,40]. Briefly, the $^3$MLCT spectrum reflects the oxidation of Ru and the associated decreased shielding of the Ru $2p$ and e$_g$ orbitals (manifested as an overall spectral shift towards higher energy, 2840 eV bleach and

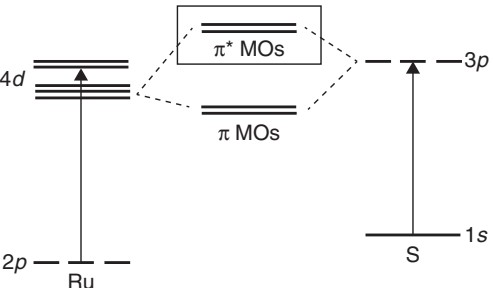

**Fig. 2** Relevant atomic orbitals probed by XAS experiments. XAS at the Ru L-edge is characterized by transitions between Ru $2p$ orbitals and molecular orbitals containing primarily Ru $4d$ character. XAS at the S K-edge is characterized by transitions between S $1s$ orbitals and molecular orbitals that are an antibonding combination of S $3p$ and Ru $4d$ orbitals ("π* MOs" indicated by solid box) and higher-energy ligand-centered orbitals

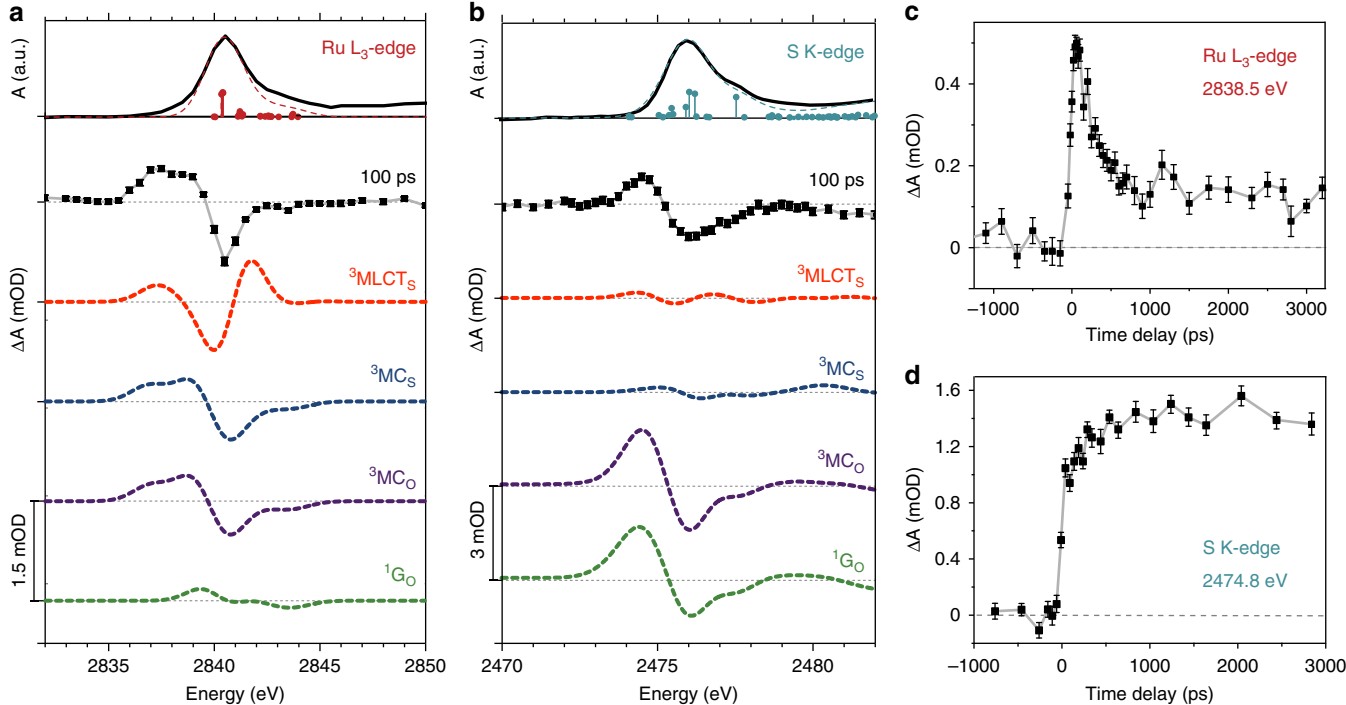

**Fig. 3** Results of TR-XAS measurements and comparison with TD-DFT simulated spectra. Measured and simulated data at the Ru $L_3$-edge (**a**) and S K-edge (**b**). Top panel (solid black trace) is the measured absorption spectrum of $[Ru(bpy)_2(pyESO)](PF_6)_2$ in the S-bonded ground state. The TD-DFT predicted transitions are indicated as sticks and the predicted spectrum is shown by the dashed line (overlaid with the measured spectrum). Below (second from top, square points) is the measured transient differential absorption spectrum at 100 ps time delay. Error bars indicate the standard error of the mean (obscured in some cases by the large size of the data points). All dashed traces below are TD-DFT simulated differential spectra of several proposed excited state intermediates (S-bonded triplet MLCT and MC states: $^3MLCT_S$ and $^3MC_S$; O-bonded MC state: $^3MC_O$) and the O-bonded ground state ($^1G_O$). The transient measured and simulated difference spectra are all plotted on the same vertical scale, with the amplitude indicated by the scale bars. **c** Fixed energy time-delay scan measured at the Ru $L_3$-edge (2838.5 eV), including error bars indicating the standard error of the mean. **d** Fixed energy time-delay scan measured at the S K-edge (2474.0 eV), with error bars indicating the standard error of the mean

2842 eV induced absorption)[35], and a low energy transition into the newly created $t_{2g}$ vacancy (see Supplementary Fig. 1). The measured differential spectrum, characterized by a narrow bleach (2.841 keV) and a single broad induced absorption at lower energies (2837–2839 eV), is inconsistent with these predicted $^3MLCT$ features.

Excited $^3MC$ states are characterized by a reconfiguration of valence electrons in the Ru $4d$ orbitals resulting in two unpaired electrons of the same spin (from $t_{2g}^6 e_g^0$ in the ground state to $t_{2g}^5 e_g^1$ in the $^3MC$ state). Therefore, the predicted $^3MC$ transient spectra (Fig. 3a) include a low energy induced absorption transition (2837 eV) to the newly created $t_{2g}$ vacancy (see Supplemental Fig. 1), and bleach (2841 eV) due to the newly occupied $e_g$ orbital. Furthermore, a predicted splitting of the $e_g$ degeneracy in the MC states contributes both to the bleach and to a second induced absorption transition (2839 eV). Comparison of the measured 100 ps transient spectrum with these predicted features provides clear evidence for $^3MC$ excited state intermediates (either $^3MC_S$ or $^3MC_O$).

The time delay scan in Fig. 3c is measured with fixed X-ray energy, resonant with the induced absorption feature (2838.5 eV) predicted for the $^3MC$ states, and the fast growth indicates that a $^3MC$ state is populated within our experimental time resolution. This observation of transient excited MC states is the first direct confirmation of previous theoretical predictions for MC-mediated isomerization pathways in Ru sulfoxide complexes[22]. This result also highlights the utility of TR-XAS to detect optically dark excited states. Although the results from previous time-resolved optical studies are consistent with MLCT-mediated isomerization

pathways[23], the insensitivity of optical transitions to MC states leads to a limited view of the reaction process. The TR-XAS results presented here enable a complete understanding of the isomerization mechanism with sensitivity to both CT and MC intermediate states.

**S K-edge TR-XAS.** In contrast to the spectra at the Ru $L_3$-edge, the orbitals probed by the S K-edge transitions are not involved in the reconfiguration of valence electrons in the MLCT or MC states. Therefore, we find that the S K-edge spectrum is sensitive only to the isomerization reaction (Fig. 3b). It is well established that the S $1s{\rightarrow}3p$ transition energy reflects the effective charge on the S atom and is therefore sensitive to its local environment[41]. For $[Ru(bpy)_2(pyESO)]^{2+}$, the change in Ru–S bonding that occurs during isomerization has a significant effect on the S K-edge spectrum. For the S-bonded ground and excited state species the Ru–S bond length changes by <3% (Supplementary Table 1) and the S atomic charge density (calculated Mulliken charge) changes by <5%. Therefore, it is not surprising that very small difference signals are predicted for the $^3MLCT_S$ and $^3MC_S$ states relative to $^1G_S$ (Fig. 3b). The O-bonded excited and ground states are characterized by Ru–S bond cleavage (>45% increases in Ru–S bond lengths, Supplementary Table 1) and an increased electron density on S that was previously donated to the Ru–S bond (Mulliken charge decreases 15%). These changes in bonding and charge result in large transient difference signals predicted for the $^3MC_O$ and $^1G_O$ states (Fig. 3b), with an overall spectral shift to lower energy (2476 eV bleach and 2474.5 eV induced absorption),

and transient difference signal amplitudes that are more than eight times larger than those predicted for the S-bonded species ($^3$MLCT$_S$ or $^3$MC$_S$). The magnitude and shape of the transient signal measured at 100 ps delay closely match the predicted spectra of the O-bonded species ($^3$MC$_O$ and $^1$G$_O$). Therefore, the transient MC intermediate species measured (identified from the Ru L$_3$-edge spectrum) can now be distinguished in terms of metal-ligand bonding (from the S K-edge spectrum) and the $^3$MC$_O$ state is clearly identified.

The time delay scan in Fig. 3d is taken at a fixed energy resonant with the induced absorption in the S K-edge transient spectrum (2474.8 eV) and consistent with formation of the O-bonded species. A two-component time response is observed, including the dominant fast increase in absorption (faster than our experimental resolution of 70 ps), indicating the formation of $^3$MC$_O$ states on this time scale. A smaller secondary increase in absorption develops on the hundreds of picosecond time scale. This suggests that O-bonded states are not exclusively formed directly from the $^3$MLCT$_S$ state, but also by another slower pathway involving a S-bonded MC state intermediate (indicated from the Ru L$_3$-edge). Optical studies also report a two-component rise time (72 ps and 640 ps) at wavelengths characteristic of the photoisomerization product[23]. Although a quantitative kinetic comparison of the two datasets is not particularly informative due to their different sensitivities (the optical absorption is attributed only to $^1$G$_O$ formation and the S K-edge difference signal is attributed to formation of all O-bonded excited and ground state species), they are consistent in reflecting the fast creation of O-bonded species. The slow rise in optical absorption was attributed to conformational relaxation of the isomerized complex; however the slow rise in S K-edge difference signal cannot be attributed to such structural changes as it is sensitive only to the Ru–S bond length which is not predicted to change significantly after formation of the $^3$MC$_O$ species (Supplementary Table 1, Supplementary Fig. 3).

The TD-DFT calculations and analysis of the potential energy landscapes reveal low transition barriers between the $^3$MLCT$_S$ surface and the $^3$MC$_S$ (1.1 kcal/mol) and $^3$MC$_O$ (1.6 kcal/mol) surfaces (Supplementary Fig. 3). The possibility to populate either $^3$MC state upon MLCT excitation explains our observation of bimodal isomerization (two-component time response in Fig. 3d) and is consistent with the predictions of transition state theory. Immediate population of the $^3$MC$_O$ state leads to fast and direct Ru—S bond dissociation, as population of an antibonding e$_g$ orbital that elongates the Ru–S bond is predicted for this state (Supplementary Table 1). However, in the $^3$MC$_S$ state a Ru–N bond elongation (orthogonal to the Ru–S bond, Supplementary Table 1) is predicted and Ru—S bond dissociation occurs only upon interconversion to the $^3$MC$_O$ state (with a 4.7 kcal/mol barrier, Supplementary Fig. 4). Interconversion between MC states with different metal-ligand bonding has also been predicted with similar transition barriers for other Ru-centered photochromic complexes[22,28]. Those predictions for a similar Ru sulfoxide complex, however, do not include the direct formation of the $^3$MC$_O$ state (from the $^3$MLCT$_S$ state), as the major structural change (Ru–S bond elongation) occurred in the $^3$MC$_S$ state and the minimum energy path connected the $^3$MLCT$_S$ and $^3$MC$_S$ states[22].

**Global analysis and isomerization kinetics**. Figure 4 presents a complete set of differential spectra at the Ru L$_3$-edge and S K-edge and corresponding time scans at fixed energy. Quantitative analysis of this complete data set provides a comprehensive picture of the isomerization pathways and intermediate states. Singular value decomposition (SVD) analysis is first applied to the Ru and S edge spectra to estimate the number of principal intermediate states that contribute significantly to the measured transient spectra. At the Ru L$_3$-edge, two major spectral components are observed (Supplementary Fig. 5) and assigned to the excited MC states ($^3$MC$_S$, $^3$MC$_O$) and the O-bonded ground state ($^1$G$_O$). These states are assigned on the basis of comparison of the individual spectral components with the simulated transient spectra of each predicted intermediate species. At the S K-edge a single spectral component is identified (Supplementary Fig. 5) and assigned to all O-bonded states ($^3$MC$_O$, $^1$G$_O$).

A proposed mechanism for photoisomerization of [Ru(bpy)$_2$(pyESO)]$^{2+}$ is shown in Fig. 5a. The data presented in the previous sections established several mechanistic requirements, which are indicated by bold arrows and include the necessity of two isomerization pathways with associated formation times for the O-bonded ground state. All other pathways included are discussed below. Global fitting of all transient spectra and fixed energy time scans at both the Ru L$_3$-edge and S K-edge was performed, based on linear combinations of the TD-DFT spectra and the kinetic equations derived for the mechanism in Fig. 5a (see Supplementary Note 1). Several variations of the photoisomerization mechanism were explored, including the addition of an O-bonded $^3$MLCT intermediate (see Supplementary Note 2 and Supplementary Fig. 6); however, the mechanism shown in Fig. 5a both provides the best fit to all data (based on least-square minimization) and satisfies all basic conditions inferred from the results presented in the previous sections. The global fitting results are overlaid with the measured differential spectra and time scans in Fig. 4. The rate constants obtained from this global fit are indicated in Fig. 5a, along with the time-dependent fractional populations of all ground and excited states in Fig. 5b.

As discussed in the previous sections, photoisomerization from the S-bonded to O-bonded species occurs through two separate pathways, and the following ratios were assigned by the global fit analysis: 40% fast formation of $^3$MC$_O$ (directly from $^3$MLCT$_S$) and 60% slower formation of $^3$MC$_O$ (from $^3$MC$_S$). An upper limit for the fast $^3$MC state formation (required to reproduce the Ru-edge transient spectra at short time delay) was established as <10 ps, as described in the Supplementary Note 3 (Supplementary Fig. 7). The sub-10 ps growth of the $^3$MC state is consistent with the previously observed ~4 ps decay of the characteristic induced absorption of MLCT states in the near-UV spectral region for this class of systems[23]. The additional slower photoisomerization pathway (required to reproduce the two-component response of the S-edge time scans) is characterized by a transition from $^3$MC$_S$ to $^3$MC$_O$ with a rate constant of 1/580 ps$^{-1}$. Subsequent relaxation from the isomerized $^3$MC$_O$ state to the O-bonded ground state ($^1$G$_O$) occurs with a rate constant of 1/430 ps$^{-1}$.

In order to account for the previously reported isomerization quantum yield, we find it necessary to introduce an additional relaxation pathway from the $^3$MC$_S$ state back to the initial S-bonded ground state (rate = 1/720 ps$^{-1}$). This does not result in an appreciable change in the S K-edge data fit quality (as the Ru–S bond is largely unchanged) and produces an isomerization quantum yield of 52%, consistent with the reported 58%[23]. In addition, a separate decay pathway from the $^3$MLCT$_S$ state back to the ground state is necessary to account for the residual MLCT signal in the Ru edge transient spectra (Supplementary Note 4 and Supplementary Fig. 8). This longer-lived $^3$MLCT state is quickly formed (within a few ps), and a decay rate of 1/170 ps$^{-1}$ to the S-bonded ground state was assigned by the global analysis. A longer-lived intermediate state (50–70 ps lifetime) with mixed $^3$MLCT and $^3$MC character has been proposed for similar Ru-centered photochromic complexes[23].

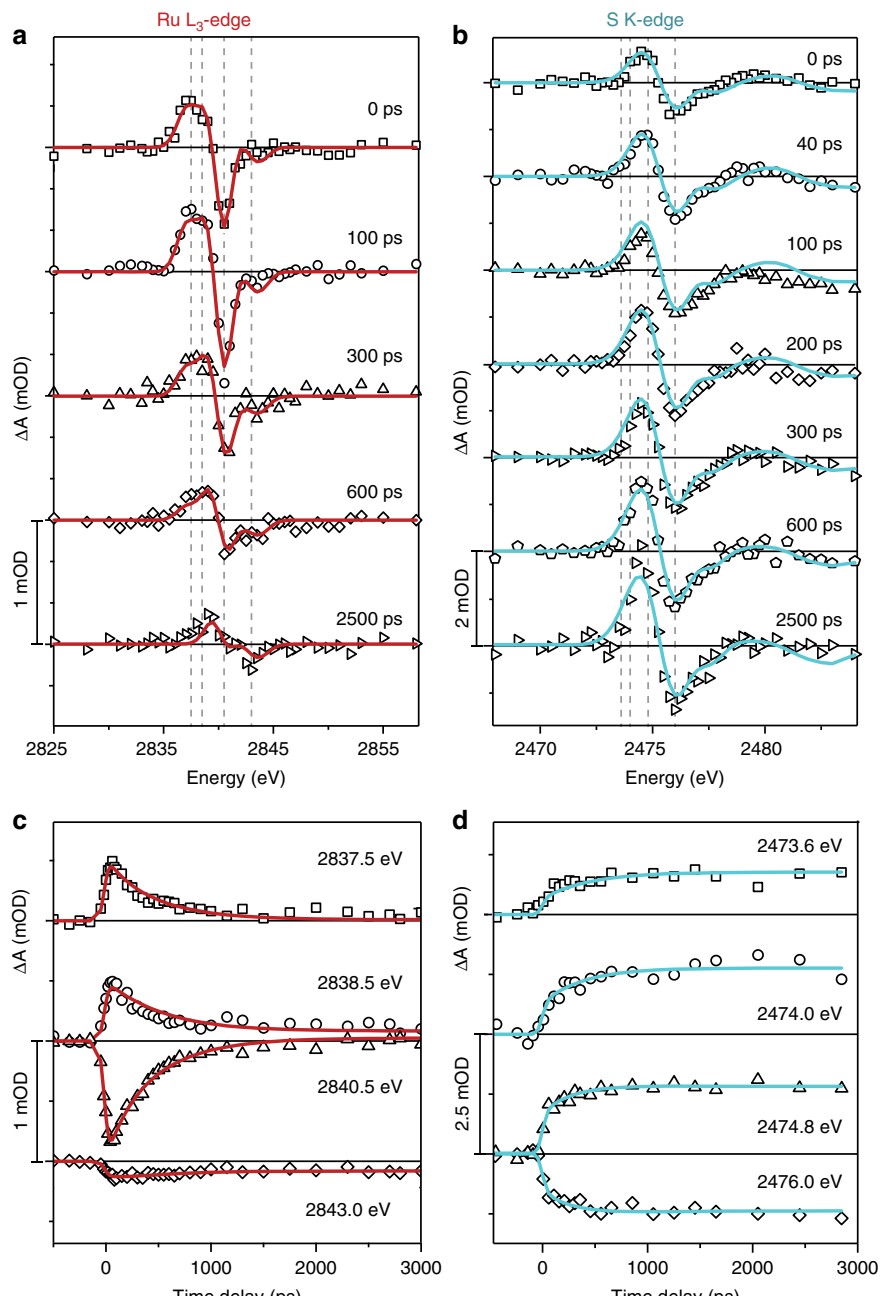

**Fig. 4** Complete TR-XAS data measured for [Ru(bpy)$_2$(pyESO)]$^{2+}$ and global fit results. Measured TR-XAS data (scattered) is overlaid with the least-square fit to the kinetic model described in the text (solid lines). **a**, **b** Differential absorption spectra at several fixed time-delays at the Ru L$_3$-edge (**a**) and S K-edge (**b**). Vertical lines indicate the energies monitored as a function of time-delay. **c**, **d** Time-delay scans at several Ru L$_3$-edge (**c**) and S K-edge (**d**) fixed energies

## Discussion

The photo-induced linkage isomerization reaction of a Ru-sulfoxide complex was followed in real-time with atomic specificity using TR-XAS and the comparison with TD-DFT simulated X-ray absorption spectra. The results highlight the unique ability of Ru L$_3$-edge XAS to distinguish CT and MC excited states with high contrast. This distinction enables the first clear experimental evidence for the critical role of long-lived MC excited states during isomerization of Ru-sulfoxide complexes, which was previously obscured by the insensitivity of optical spectroscopy to MC states. Furthermore, these results demonstrate that the S K-edge spectrum is primarily sensitive to the Ru–S bond length. Therefore, the combined approach of metal and ligand atom XAS

enables the correlation of electronic configuration and Ru–S bonding required to determine the detailed isomerization reaction mechanism presented here.

Photoisomerization of [Ru(bpy)$_2$(pyESO)]$^{2+}$ is found to proceed through two reaction pathways, both of which find the initially formed $^3$MLCT$_S$ excited state depopulated by coupling with $^3$MC states. The antibonding nature of the $^3$MC states facilitate the structural changes that make the linkage isomerization possible. One reaction pathway likely proceeds through a dissociative state (S-bonded $^3$MC) with an elongating Ru–S bond (reminiscent of a previously calculated linkage isomerization for a similar Ru–sulfoxide complex)[22], resulting in fast formation of the O-bonded $^3$MC$_O$ state. A second pathway first

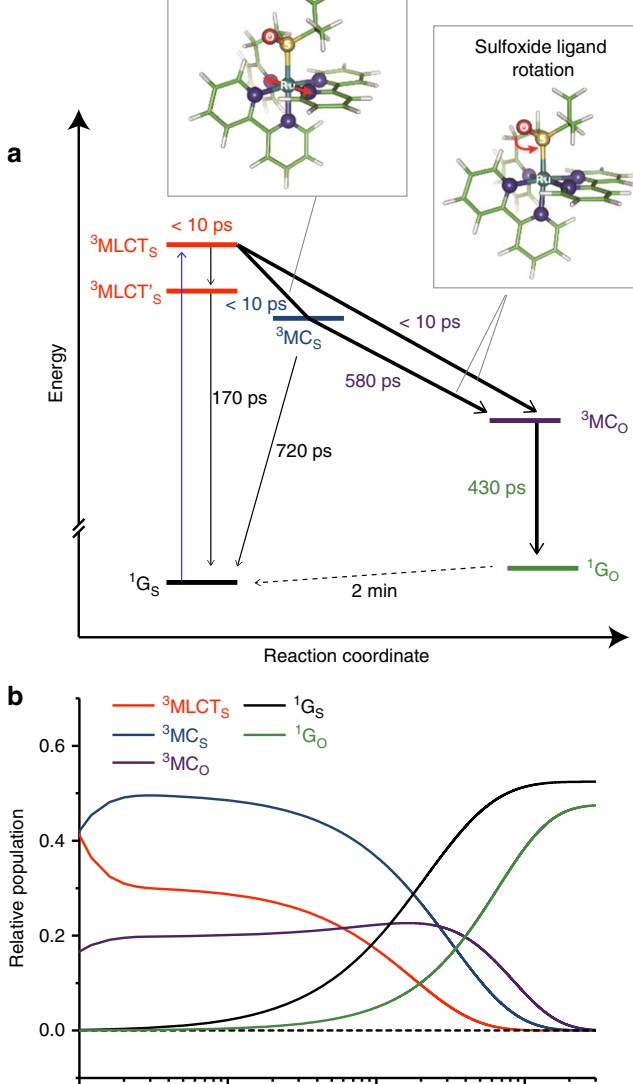

**Fig. 5** Proposed mechanism and kinetics for the photoisomerization of [Ru(bpy)$_2$(pyESO)]$^{2+}$. **a** Proposed photoisomerization mechanism with labels indicating electronic state and lifetime for each ground state and intermediate structure. Bold arrows indicate the basic mechanistic requirements required to describe the Ru and S edge data, including the necessity of MC excited states (Ru edge) and of two isomerization pathways (S edge), and the formation time for the O-bonded ground state. Insets show the major structural changes that accompany electronic transitions. **b** Time-dependent fractional population for each reaction intermediate

elongates the Ru–N bonds perpendicular to the Ru–S bond ($^3$MC$_S$ state), forming the O-bonded $^3$MC$_O$ state more slowly. The formation of such MC states generally facilitates isomerization and dissociation reactions in transition metal complexes, as they are accompanied by larger structural rearrangements than CT states due to the occupation of antibonding molecular orbitals aligned with the metal-ligand bonds[42]. Therefore, the population of MC excited states can facilitate the efficiency of molecular devices through directed motion or through the creation of reactive species, in the cases of ligand isomerization and dissociation, respectively.

For the class of Ru-based photochromic complexes containing sulfoxide ligands, the proposed role of MC excited states has varied. For chelating sulfoxide ligands, such as pyESO, high isomerization quantum yields have been attributed to non-adiabatic transitions between the triplet excited state and singlet ground state PESs. The potential energy crossing points were predicted by DFT to occur around the $^3$MC state minima[22,28,29], highlighting the role of MC states in facilitating efficient isomerization. However, the optical spectroscopy suggested that large structural changes occur in the CT excited states and the crossing points were predicted to occur around the $^3$MLCT minima[20,21,23–25]. In this work we have demonstrated that the high isomerization quantum yield of [Ru(bpy)$_2$(pyESO)]$^{2+}$ is achieved mainly by fast adiabatic formation of the $^3$MC$_O$ state from the $^3$MLCT$_S$ state. Although non-adiabatic transitions may play a role in forming the Ru–O bond in the ground state (Supplementary Fig. 4), the major structural changes associated with ligand isomerization occur through formation of the $^3$MC$_O$ state adiabatically on the triplet PES. Studies like the one presented here, that can simultaneously resolve both the electronic configuration of the central metal atom and metal-ligand bonding, are essential to determine the precise role of MC states during photoisomerization and how that role depends on the nature of the isomerizing ligand.

## Methods

**Experimental procedures**. X-ray absorption measurements were performed at the time-resolved beamline 6.0.1 of the Advanced Light Source (ALS) at Lawrence Berkeley National Laboratory. Synthesis of [Ru(bpy)$_2$(pyESO)](PF$_6$)$_2$ followed published procedures[23]. The sample was dissolved in propylene carbonate (15 mM), which was continuously refreshed using a sapphire nozzle liquid jet (50 um thickness) in a He atmosphere. The undulator beamline is equipped with a chopper running at 4 kHz, thus reducing the average X-ray flux and determining the data acquisition rate. A double crystal Si(111) monochromator produced monochromatic X-rays tunable around the Ru L$_3$-edge (~2840 eV) and S K-edge (~2470 eV) with 0.35 eV resolution. Two ALS filling modes were used: "multibunch" for the Ru L$_3$-edge (276 electron bunches spaced by 2 ns and one so-called "camshaft" bunch separated by a 100 ns filling gap), and "two-bunch" mode for the S K-edge (two "camshaft" bunches separated by 328 ns). In both filling modes the X-ray pulses are 70 ps (FWHM) in duration. A fast avalanche photodiode was used to gate and record one X-ray "camshaft" pulse at 4 kHz after transmission through the liquid jet. The X-ray spot size was $180 \times 160$ μm$^2$ at the sample position.

The optical pump was produced by an amplified Ti:Sapphire laser system described elsewhere[35]. The laser frequency was down-converted to 2 kHz (to interleave X-ray and laser pulses) and the 400 nm pump wavelength was derived by second harmonic generation. The pump wavelength was selected in order to be close to the peak of the absorption feature attributed to MLCT excitation (see optical absorption spectrum in ref. [23]). The pump pulse was temporally stretched to ~500 fs using fused silica. The laser overfilled the X-ray spot size and the pump pulse fluence was 100 mJ/cm$^2$. Laser and X-ray pulses were overlapped at the sample spatially and temporally using a pinhole and fast APD detector, respectively.

Custom electronics were used to synchronize the laser pulses to the ALS orbit clock and chopper openings, thus allowing precise control of the time delay between laser and X-ray pulses. The transmitted X-ray intensity was collected with and without the laser pump as a function of monochromator energy or pump-probe time delay. Differential absorption ($\Delta A$) was calculated as

$$\Delta A = -\log\left(\frac{I_{on}}{I_{off}}\right)$$

where $I_{on}$ and $I_{off}$ are transmitted intensity with and laser excitation, respectively.

**Theoretical calculations**. All reaction species of [Ru(bpy)$_2$(pyESO)]$^{2+}$ were calculated using DFT in the gas phase using the program Gaussian 09[43]. All possible geometries were fully optimized using the hybrid functional of Perdew, Burke, and Ernzerhof (PBE0)[44] combined with def2-SVP basis set[45] for all atoms. To treat the scalar relativistic effect for Ru atoms, we employed the relativistic effective core potential (RECP)[46]. All transition states were identified by one imaginary frequency and confirmed by the intrinsic reaction coordinate (IRC) method[47,48]. The reaction PES (Supplementary Fig. 4) reflects the zero-point corrected energies for each intermediate state, relative to the S-bonded ground state. These energies were re-calculated using the polarizable continuum model to account for the propylene carbonate solvent environment present in the experiments (Supplementary Table 2); however, the solvent did not affect the relative energies of the intermediate states and was not considered further.

The XANES spectra of Ru $L_3$- and S K-edges were simulated from the optimized geometries described above with TD-DFT calculations using the ORCA 2.9.1 program[49]. The PBE0 functional was implemented[44]. Scalar relativistic effects and solvent effects were addressed using the second-order Douglas–Kroll–Hess Hamiltonian (DKH2)[50] and conductor-like screening model (COSMO)[51], respectively. The Tamm–Dancoff approximation (TDA)[52] and the resolution of the identity Coulomb with the chain of spheres exchange algorithms (RIJCOSX)[53,54] were employed for the efficiency of calculations. The segmented all-electron relativistically contracted (SARC) def2-TZVP(-f) basis set[55] and decontracted def2-TZVP/J auxiliary basis set for all atoms were used. The tight SCF convergence criteria and DFT integration grid (Grid4) were implemented to obtain a reliable accuracy of calculation. The absolute energy depends on the computational functionals and basis sets, and shifts of −5.45 eV and 47.80 eV were required to match the experimental Ru $L_3$-edge and S K-edge spectra, respectively, which are of reasonable magnitudes based on previous work[36,50]. A Gaussian broadening of 1.6 eV was used to match the calculated and measured spectral peak width.

**Data availability**. The data that support the findings of this study are available from the corresponding authors upon reasonable request.

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

## Acknowledgements

This work was supported by the U.S. Department of Energy, Office of Science, Basic Energy Sciences, Chemical Sciences, Geosciences, and Biosciences Division. This research used resources of the Advanced Light Source, Lawrence Berkeley National Laboratory, which is a DOE Office of Science User Facility. T.K.K. acknowledges funding from the National Research Foundation (NRF) grants, funded by the Ministry of Science and ICT (No. 2016K2A9A1A01945137, 2016R1E1A1A01941978, 2014R1A4A1001690, and 2016K1A4A4A01922028). N.H. acknowledges funding from the Max Planck Society, the City of Hamburg, and the German Science Foundation (DFG) through the SFB 925 "Light induced dynamics and control of correlated quantum systems". J.J.R. and K.G. acknowledge funding from the National Science Foundation (CHE 0947031, CHE 1112250, and CHE 1602240).

## Author contributions

J.H.L., K.H., A.A.C, H.C., N.H., T.K.K., and R.W.S. designed and carried out the TR-XAS experiments. K.G. and J.J.R. provided the sample. K.H. and J.H.L., advised by M.B.P., carried out DFT calculations. J.H.L. provided the kinetic analysis. All authors contributed to the analysis of results and manuscript text.

## Additional information

**Competing interests:** The authors declare no competing interests.

