## [Peer Review File · Nature Communications]

Reviewers' comments:

Reviewer #1 (Remarks to the Author):

The authors studied the photoisomerization reaction of a Ru-based complex in solution. The manuscript reports the results of time-resolved x-ray absorption spectroscopy measurements (at the Ru L and the S K absorption edges) and time-dependent density-functional theory calculations. The major claim of these investigations is that hitherto undetected metal-centered intermediate states "mediate" (see title) the isomerization reaction.

Light-triggered isomerization reactions are ubiquitous in nature. A mechanistic understanding of ultrafast photoisomerization reactions with no doubt represents a formidable challenge as the potential impact, control of chemical (isomerization) reactions based on rational design, is enormous. For this the authors chose a, nowadays, classical approach by addressing a well-defined, specifically made and previously well studied metal-centered model complex (see the previous publications by the Rack and Boggio-Pasqual groups, co-authors of the current manuscript).

This is the setting which enables the first and probably most important novelty of the current manuscript: The chosen experimental approach of time-resolved x-ray absorption spectroscopy impressively allows the now joined teams of authors (adding the x-ray spectroscopy groups of Schoenlein and Kim) to fill the essential knowledge gaps concerning the system studied. By probing with (sufficient) temporal resolution (of around 100 ps) and, more importantly, with elemental specificity the metal and one of the ligands involved in the linkage isomerization reaction (S) the authors can impressively ascertain the involvement of theoretically proposed metal-centered states in the reaction. "The contrast" with which this can be done here (distinction of 3MLCT and 3MC states with Ru L-edge spectra and distinction of 3MC states with S or O bound to Ru) is very impressive (with a reservation about the S K-edge spectra potentially reducing the validity, see below).

In this respect and based on this objective assessment, this study is an excellent example for how the comparably uncommon method of time-resolved x-ray spectroscopy can be used to add essential and otherwise inaccessible information for the study of ultrafast chemical reactions. Because the finding is so "clear-cut" the presented results could become a benchmark in the field of chemical reaction dynamics studies with time-resolved x-ray spectroscopy. Clearly, the unique x-ray spectroscopic results inherently render the conclusions original.

In addition, the result in terms of reaction mechanism is the more impressive as the numerous previous studies by part of the authors of the manuscript well prepared the ground but clearly left open essential questions about the reaction mechanism due to the inherent lack of the previously employed experimental methods. Detection and characterization of short-lived intermediate states in chemical reaction dynamics is obviously essential for concluding on the reaction mechanism. The unambiguousness in the characterization of intermediates and the corresponding conclusion on the reaction mechanism reached here defines the large extent to which this study should be of interest to the wider field of chemical reaction dynamics in general and to the field of ultrafast linkage isomerization reactions specifically.

Two aspects, however, may reduce the potential impact of the current manuscript now looking at it from a more subjective perspective. First, information seems to be missing in order to clarify in which way the results will influence thinking in the field. Second, additional information seems necessary to eliminate remaining doubts about the data analysis and to further ascertain the validity of the claims. These are detailed below.

Thinking in the field:

The discussion of the current manuscript (p. 16/17) falls short of the importance and impact of the main claim. Instead of placing it into context, the discussion merely summarizes the finding, including repetitions. The major finding is reported twice (on p. 16 "Photoisomerization ... is found to proceed through two adiabatic transitions, forming the 3MCO state from either the 3MLCTS or 3MCS intermediate states" and on p. 17 "the results presented here indicate that the major structural changes associated with ligand isomerization occur adiabatically, on the triplet potential energy surface"). The text around these summarizing statements seems rather descriptive and the last paragraph of the discussion again summarizes the results. It is very clearly stated how the novel results have been achieved and this is excellent. But the authors missed the chance to increase the impact of the results by placing them into context with other results on other linkage isomerization or photoisomerization reactions. In order not to make the results look small because they seem to report "some detail on yet another molecule" this discussion could or should include comparisons to other complexes, possible implications of the claim in terms of how we think about linkage isomerization reactions of this or comparable complexes. This may be speculative but necessary. Rudimentary, this is done in the present manuscript (p. 16 "Here we suggest the high isomerization quantum yield observed for $[\text{Ru}(\text{bpy})_2(\text{pyESO})]^{2+}$ is achieved mainly by fast formation of the 3MCO state, following the adiabatic transition from the 3MLCTS state." and p. 17 "non-adiabatic transitions play a role in forming the Ru-O bond in the ground state, the results presented here") and it could be sufficient to extend these arguments.

In the end the question remains: Why is the claimed "critical role of long-lived MC excited states during isomerization of Ru-sulfoxide complexes" such an important finding? What does it or could it mean when "transition metal-centered states mediate isomerization" (see title)? This directly relates to the "essential open question in functional transition-metal complexes" the authors pose in the introduction about "the relative roles of transient charge-transfer states ... versus metal-centered (MC) excited states ... in mediating the quantum efficiency of photo-induced processes such as isomerization". This question seems to remain open. More thoughts would be highly appreciated by the readership, could stimulate further research and enhance impact of the study.

Validity of the claims:

In order to better assess the way the electronic structure is probed here it would be valuable to have a molecular-orbital diagram to connect the transitions in Figure 1b with bonding in the complex. By just indicating the dipole-allowed (hence obvious) 2p-4d and 1s-3p transitions, essential links between the spectroscopy and the electronic structure (bonding) remain elusive. It remains unclear, e.g., what the S 3p orbitals interact with in the given complex. One may therefore ask why orbital the population (of the π^* e.g.) cannot be seen in the S K-edge spectra. This may, however, also be a misunderstanding given that fact that the molecular-orbital diagram is not given. It seems important to have it in the current manuscript to better link the reported x-ray spectroscopic results with common and previous knowledge about this system.

The intensities of the calculated spectra are dubious. Strictly speaking, if intensities are plotted in arbitrary units such as in the current version of Figure 2, they could or should be all normalized to the same intensity at maximum (or minimum). The same problem of seemingly meaningful but obscure relative intensities appears in Figure 3 (all panels). In any case and, again, strictly speaking, no conclusions should be drawn from intensities in arbitrary units. This is important this challenges one of the main arguments concerning "contrast" between 3MLCTS, 3MCS and 3MCO S K-edge spectra (Figure 2b): Why are the intensities of the 3MLCTS and 3MCS spectra so small? They show oscillations but these are not discussed. Is there information connected to the intensity of the spectra (relative intensities of difference spectra)? Why were the spectra plotted with different intensities anyway? It seems necessary that the authors provide clearer evidence for why the intensities are small and the contributions of the respective species are hence considered irrelevant. This may be possibly explained with the information given in the Supplementary Information. However, unless this is really clarified, claims about distinguishing 3MLCTS, 3MCS

and 3MCO states appear not to be substantiated (such claims include the following: p. 7: "from the perspective of the S site (Fig. 2b), the TD-DFT predicted differential spectra for the 3MCS and 3MCO states are clearly distinguishable both in spectral shape and amplitude", p. 7/8: "It is clear from visual comparison (Fig. 2a) that the differential spectrum measured at 100 ps time delay does not match the simulated MLCT state spectrum.", p. 10 "The simulated transient spectra for the S-bonded intermediates (both 3MLCTS and 3MCS) have very small transient signals...").

To further enhance validity of the claims (in particular any discussion of the intensity minimum in the differential R L-edge spectra at around 2843 eV or the "smaller secondary increase in absorption develops on the hundred picosecond time scale" in Figure 2d) it seems necessary to include error bars in the experimental data in Figure 2a and 2b. This may not be necessary in Figure 3 where experimental and calculated data are directly overlaid.

Time scales (time constants) extracted from the current measurements (fit of kinetic model to the data) appear not to be compared to the previous optical data. Is there a specific reason for this? Do they match or do they not match and what would it mean if they did not match?

The authors nicely connect the charge on Ru/S with spectral shapes/shifts (p. 9: "In the MC states, the overall electron density on the Ru atom is similar to the ground state.", p. 10: "The $1s\sigma 3p$ transition energy reflects the effective charge on the S atom..."). Where is the evidence for the claims of how charges (charge densities) change or do not change? The references given seem to describe very different complexes. Have local charge densities been calculated? Without quantifying these claims (albeit accepting the possible uncertainties in calculating charge densities in metal complexes) the ensuing conclusions or interpretations seem unsubstantiated. The exemplary and important sentence that illustrates the need to validate claims made with respect to the interpretation of the x-ray absorption spectra by providing more information on the actual charge density on the respective atoms and its relation to the energies of the final (core-excited) states is on p. 8/9: "The predicted 3MLCTS transient spectrum includes an overall spectral shift towards higher energy ... reflecting the Ru 2p core level shift that results from removal of a Ru 4d electron." How can the authors be certain that the full equivalent of one Ru 4d electron is removed from the Ru atom? How can they be certain that this is the only cause of the spectral shift? This may relate to the statement on p. 11 "the transient signal is not predicted to grow in magnitude upon any structural changes occurring after formation of the 3MCO state" which seems unsubstantiated as no evidence (spectra?) seems to be given.

All discussions of spectral shifts are made in a one-electron orbital-based picture. A priori, this is often too simplified (see e.g. the spin-orbit interaction for Ru L-edge final states). Multiplet effects may be agreeably small for Ru L-edge spectroscopy but evidence has to be provided to support the one-electron picture a posteriori (to validate the approximations made implicitly). This could be done by linking better the main text to the Supplementary Information (Figs. S1 and S2). In this respect: What exactly is meant with the "Binding Energy" in Figure 1b? If it was the orbital energy it would have to increase from bottom to top, if it was minus the orbital energy, would the arrow not have to be inverted? What is meant with the dashed line?

In order to better assess validity of the calculated differential spectra it would be good to see the comparison of the calculated and measured ground-state spectra (in the Supplementary Information, e.g.).

Supplementary Information:

Figures S1 and S2: Transitions are very hard to assess because line spectra are given with squares that seem to indicate the transitions. Sticks or lines would be much more useful indicating the transitions.

Reviewer #2 (Remarks to the Author):

The authors investigated electronic structures of Photochromic Ru-Sulfoxide Complex during its photoinduced isomerization processes where a Ru-S bond was transformed to a Ru-O bond accompanied by internal rotation of the S-O group as well as other parts of the molecule. Compared to previous optical studies that have been carried out by Rack's group who also synthesized the compounds, the new development is the use of TR-XAS measurements at Ru L-edge and S K-edge which are sensitive to the electronic structural changes complementary to what can be obtained from optical transient absorption measurements. They also carried out theoretical calculations to establish the correlations between the observed spectral changes with the electronic structural changes of this complex. From this study, they extracted a more detailed reaction pathways with different states which otherwise will not be extracted from previous studies.

The study has been carried out carefully, but some issues need to be addressed before it can be reconsidered as listed below.

1. The photochemical theme is unclear until the discussion part, which makes the results hard to follow with their connections to the reaction mechanisms. It at least should draw out the initially hypothesized reaction mechanisms and all the states involved at the beginning. Perhaps a reaction scheme with all the possible states involved and their presumed energy levels will help. Of course, it may run into the conflict in the discussion section where the authors currently displaced such a diagram to summarize the results.

2. The authors brought up the adiabatic and non-adiabatic reaction mechanism for the isomerization process, which is very interesting to consider. However, describing such mechanisms normally requires some excited state potential energy surfaces. It is unclear what the adiabatic and non-adiabatic reaction mechanisms are defined here, and which potential energy parabola are involved. Such depiction could help to clarify the confusion. How should readers understand the non-adiabatic process that is correlated with the isomerization. It seems that the authors should clarify two sets of parameters, the pathway along the isomerization involving the nuclear motions and the energetic pathway. If they can do that the significant of the manuscript will be much higher.

3. The X-ray excitation diagram in Figure 1 seems unnecessary because that belongs to the definitions of L- and K-edge transitions shown in the textbook.

4. I would not agree with their statement: "L-edge spectroscopy probes transition-metal d orbitals, while K-edge spectroscopy probes ligand atom p orbitals". It is a simple-minded description and could cause misleading in the community. The authors should add L-edge Mainly probes..." to be more accurate because the pre-edge in K-edge spectra could also provide information for d as observed in many studies.

5. It seems that an optical absorption spectrum in the UV/vis region will be good for readers to appreciate the work better.

6. Because of the lack for introducing different states, it is hard to read time-evolution difference spectra in Figure 2. Why there are difference spectral signals for the ground state G, or if G stands for the ground state? Definitions are important.

7. P. 9 what is the "intermediate MC state"?

In summary, I would not recommend the publication of this manuscript in Nature Communication at present form without further revision as outlined above. However, the manuscript can be revised from the excellent work done by this team for further discussions.

Response to review comments

Reviewer #1:

The authors studied the photoisomerization reaction of a Ru-based complex in solution. The manuscript reports the results of time-resolved x-ray absorption spectroscopy measurements (at the Ru L and the S K absorption edges) and time-dependent density-functional theory calculations. The major claim of these investigations is that hitherto undetected metal-centered intermediate states “mediate” (see title) the isomerization reaction.

Light-triggered isomerization reactions are ubiquitous in nature. A mechanistic understanding of ultrafast photoisomerization reactions with no doubt represents a formidable challenge as the potential impact, control of chemical (isomerization) reactions based on rational design, is enormous. For this the authors chose a, nowadays, classical approach by addressing a well-defined, specifically made and previously well studied metal-centered model complex (see the previous publications by the Rack and Boggio-Pasqual groups, co-authors of the current manuscript).

This is the setting which enables the first and probably most important novelty of the current manuscript: The chosen experimental approach of time-resolved x-ray absorption spectroscopy impressively allows the now joined teams of authors (adding the x-ray spectroscopy groups of Schoenlein and Kim) to fill the essential knowledge gaps concerning the system studied. By probing with (sufficient) temporal resolution (of around 100 ps) and, more importantly, with elemental specificity the metal and one of the ligands involved in the linkage isomerization reaction (S) the authors can impressively ascertain the involvement of theoretically proposed metal-centered states in the reaction. “The contrast” with which this can be done here (distinction of 3MLCT and 3MC states with Ru L-edge spectra and distinction of 3MC states with S or O bound to Ru) is very impressive (with a reservation about the S K-edge spectra potentially reducing the validity, see below).

In this respect and based on this objective assessment, this study is an excellent example for how the comparably uncommon method of time-resolved x-ray spectroscopy can be used to add essential and otherwise inaccessible information for the study of ultrafast chemical reactions. Because the finding is so “clear-cut” the presented results could become a benchmark in the field of chemical reaction dynamics studies with time-resolved x-ray spectroscopy. Clearly, the unique x-ray spectroscopic results inherently render the conclusions original.

In addition, the result in terms of reaction mechanism is the more impressive as the numerous previous studies by part of the authors of the manuscript well prepared the ground but clearly left open essential questions about the reaction mechanism due to the inherent lack of the previously employed experimental methods. Detection and characterization of short-lived intermediate states in chemical reaction dynamics is obviously essential for concluding on the reaction mechanism. The unambiguousness in the characterization of intermediates and the corresponding conclusion on the reaction mechanism reached here defines the large extent to which this study should be of interest to the wider field of chemical reaction dynamics in general and to the field of ultrafast linkage isomerization reactions specifically.

Two aspects, however, may reduce the potential impact of the current manuscript now looking at it from a more subjective perspective. First, information seems to be missing in order to clarify in which way the results will influence thinking in the field. Second, additional information seems necessary to eliminate remaining doubts about the data analysis and to further ascertain the validity of the claims. These are detailed below.

Thinking in the field:

The discussion of the current manuscript (p. 16/17) falls short of the importance and impact of the main claim. Instead of placing it into context, the discussion merely summarizes the finding, including repetitions. The major finding is reported twice (on p. 16 “Photoisomerization ... is found to proceed through two adiabatic

transitions, forming the 3MCO state from either the 3MLCTS or 3MCS intermediate states” and on p. 17 “the results presented here indicate that the major structural changes associated with ligand isomerization occur adiabatically, on the triplet potential energy surface”). The text around these summarizing statements seems rather descriptive and the last paragraph of the discussion again summarizes the results. It is very clearly stated how the novel results have been achieved and this is excellent. But the authors missed the chance to increase the impact of the results by placing them into context with other results on other linkage isomerization or photoisomerization reactions. In order not to make the results look small because they seem to report “some detail on yet another molecule” this discussion could or should include comparisons to other complexes, possible implications of the claim in terms of how we think about linkage isomerization reactions of this or comparable complexes. This may be speculative but necessary. Rudimentary, this is done in the present manuscript (p. 16 “Here we suggest the high isomerization quantum yield observed for [Ru(bpy)₂(pyESO)]²⁺ is achieved mainly by fast formation of the 3MCO state, following the adiabatic transition from the 3MLCTS state.” and p. 17 “non-adiabatic transitions play a role in forming the Ru-O bond in the ground state, the results presented here”) and it could be sufficient to extend these arguments.

In the end the question remains: Why is the claimed “critical role of long-lived MC excited states during isomerization of Ru-sulfoxide complexes” such an important finding? What does it or could it mean when “transition metal-centered states mediate isomerization” (see title)? This directly relates to the “essential open question in functional transition-metal complexes” the authors pose in the introduction about “the relative roles of transient charge-transfer states ... versus metal-centered (MC) excited states ... in mediating the quantum efficiency of photo-induced processes such as isomerization”. This question seems to remain open. More thoughts would be highly appreciated by the readership, could stimulate further research and enhance impact of the study.

Response 1:

The entire discussion has been rewritten in order to better frame these new results in the broader context of photoisomerization reactions and chemical reaction dynamics. In particular, the discussion section now better highlights the following two main points:

1. The benchmark nature of this work in providing a method capable of resolving MC and CT excited states with high contrast. The combination of metal and ligand atom spectroscopies, sensitive to the electronic configuration and bonding of the isomerizing ligand, respectively.
2. The role of MC states and how they are expected to influence the photochemistry of molecular devices more generally is now addressed. The proposed/presumed role of MC states has varied for Ru sulfoxide complexes in previous work, without definitive evidence. This now discussed in more detail (also highlighted in the introduction) and compared to the findings of this work in which the precise role of MC states is determined by resolving both the electronic configuration of the central metal atom and metal-ligand bonding during the reaction.

Validity of the claims:

In order to better assess the way the electronic structure is probed here it would be valuable to have a molecular-orbital diagram to connect the transitions in Figure 1b with bonding in the complex. By just indicating the dipole-allowed (hence obvious) 2p-4d and 1s-3p transitions, essential links between the spectroscopy and the electronic structure (bonding) remain elusive. It remains unclear, e.g., what the S 3p orbitals interact with in the given complex. One may therefore ask why orbital the population (of the pi* e.g.) cannot be seen in the S K-edge spectra. This may, however, also be a misunderstanding given that fact that the molecular-orbital diagram is not given. It seems important to have it in the current manuscript to better link the reported x-ray spectroscopic results with common and previous knowledge about this system.

Response 2:

A MO diagram (Figure 2) has been added to illustrate the Ru and S bonding interactions. The text has also been modified to better connect to the illustrations of the molecular orbitals probed in this work, shown in Supplemental Figures 1-2.

The first paragraph of the Results section now reads:

“The Ru L-edge spectrum is comprised of transitions from 2p core levels to unoccupied orbitals of primarily Ru 4d character (dx^2-y^2 and dz^2), as illustrated in Supplementary Fig. 1. The S K-edge spectrum is comprised of transitions from the S 1s core level to antibonding orbitals of primarily sulfoxide ligand character (with some metal character contributing to the lower energy peak), as illustrated in Supplementary Fig. 2”

The intensities of the calculated spectra are dubious. Strictly speaking, if intensities are plotted in arbitrary units such as in the current version of Figure 2, they could or should be all normalized to the same intensity at maximum (or minimum). The same problem of seemingly meaningful but obscure relative intensities appears in Figure 3 (all panels). In any case and, again, strictly speaking, no conclusions should be drawn from intensities in arbitrary units. This is important this challenges one of the main arguments concerning “contrast” between 3MLCTS, 3MCS and 3MCO S K-edge spectra

Response 3:

The reviewer has brought up an important point, which was in need of clarification in the manuscript. The units of intensity should not have been reported as “arbitrary”, as the intensities reflect the transition cross-sections (both measured and calculated) or the change in cross-section (in the case of transient signals). Intensity scale bars and units have been added to Figure 3 in the revised manuscript.

The relative intensities of the difference signals shown in Figure 3 are significant, as they indicate the amount of spectral change relative to the S-bonded ground state. The scaling of the simulated difference spectra are now addressed more explicitly in the main text, which has been revised to state:

“The predicted differential spectra shown in Fig. 3 were scaled to fit the 1G_0 predicted differential spectrum to that measured after completion of the photoisomerization reaction (at time delay = 2.5 ns).”

The description of the global fit in the SI has also been revised to address this point and now reads “The fitting parameters included the rate constants from the rate equations above, the time delay corresponding to temporal overlap of X-ray and laser pulses, the X-ray temporal width, and scaling factors between the theoretical and experimental data (scaling of theory to experiment is done separately for Ru and S datasets).”

(Figure 2b): Why are the intensities of the 3MLCTS and 3MCS spectra so small? They show oscillations but these are not discussed. Is there information connected to the intensity of the spectra (relative intensities of difference spectra)? Why were the spectra plotted with different intensities anyway? It seems necessary that the authors provide clearer evidence for why the intensities are small and the contributions of the respective species are hence considered irrelevant. This may be possibly explained with the information given in the Supplementary Information. However, unless this is really clarified, claims about distinguishing 3MLCTS, 3MCS and 3MCO states appear not to be substantiated (such claims include the following: p. 7: “from the perspective of the S site (Fig. 2b), the TD-DFT predicted differential spectra for the 3MCS and 3MCO states are clearly distinguishable both in spectral shape and amplitude”,

Response 4:

The relative intensities of the S K-edge difference spectra are significant and are now addressed more thoroughly in the manuscript. Briefly, the spectral changes that result in significant difference

intensities only occur upon a major change in the Ru-S interaction. This is now addressed in detail in the manuscript.

Page 11 paragraph 3 now reads:

“For $[\text{Ru}(\text{bpy})_2(\text{pyESO})]^{2+}$, the change in Ru-S bonding that occurs during isomerization has a significant effect on the S K-edge spectrum. For the S-bonded ground and excited state species the Ru-S bond length changes by <3% (Supplementary Table 1) and the S atomic charge density (calculated Mulliken charge) changes by <5%. Therefore, it is not surprising that very small difference signals are predicted for the ${}^3\text{MLCT}_\text{S}$ and ${}^3\text{MC}_\text{S}$ states relative to ${}^1\text{G}_\text{S}$ (Fig. 3b). The O-bonded excited and ground states are characterized by Ru-S bond cleavage (>45% increases in Ru-S bond lengths, Supplementary Table 1) and an increased electron density on S that was previously donated to the Ru-S bond (Mulliken charge decreases 15%). These changes in bonding and charge result in large transient difference signals predicted for the ${}^3\text{MC}_\text{O}$ and ${}^1\text{G}_\text{O}$ states (Fig. 3b), with an overall spectral shift to lower energy (2476 eV bleach and 2474.5 eV induced absorption), and transient difference signal amplitudes that are more than eight times larger than those predicted for the S-bonded species (${}^3\text{MLCT}_\text{S}$ or ${}^3\text{MC}_\text{S}$). The magnitude and shape of the transient signal measured at 100 ps delay closely match the predicted spectra of the O-bonded species (${}^3\text{MC}_\text{O}$ and ${}^1\text{G}_\text{O}$).”

p. 7/8: “It is clear from visual comparison (Fig. 2a) that the differential spectrum measured at 100 ps time delay does not match the simulated MLCT state spectrum.”,

Response 5:

This comment refers to the Ru L-edge data, in which it is clear from visual comparison that the spectral shape of the transient data is inconsistent with the simulated MLCT difference spectrum (the large induced absorption at high energy simulated for the MLCT difference spectrum is not present in the transient data). This statement is unrelated to the relative intensities, having more to do with the spectral shape.

p. 10 “The simulated transient spectra for the S-bonded intermediates (both ${}^3\text{MLCT}_\text{S}$ and ${}^3\text{MC}_\text{S}$) have very small transient signals...”).

See response 4.

To further enhance validity of the claims (in particular any discussion of the intensity minimum in the differential R L-edge spectra at around 2843 eV or the “smaller secondary increase in absorption develops on the hundred picosecond time scale” in Figure 2d) it seems necessary to include error bars in the experimental data in Figure 2a and 2b. This may not be necessary in Figure 3 where experimental and calculated data are directly overlaid.

Response 6:

Error bars have been added to Figures 3a-d (though they are somewhat obscured by the large data points used for the differential spectra in Figures 3a and 3b).

Time scales (time constants) extracted from the current measurements (fit of kinetic model to the data) appear not to be compared to the previous optical data. Is there a specific reason for this? Do they match or do they not match and what would it mean if they did not match?

Response 7:

Several of the conclusions of this work are consistent with features of the previously reported optical spectroscopy. However, quantitative comparison of the time constants extracted from the respective data sets is not particularly informative since the optical spectroscopy is not sensitive to the MC excited states. Nevertheless, we have added the following comparisons of the two datasets:

1. Optical TA found that the induced absorption characteristic of an MLCT excited state for bipyridyl complexes (around 360 nm) decays in ~4 ps.

Page 16, paragraph 2 now reads:

“The sub-10 ps growth of the ^3MC state is consistent with the ~ 4 ps decay of the characteristic induced absorption of MLCT states in the near-UV spectral region for this class of systems.²³”

2. The optical induced absorption at 480 nm is consistent with the formation of the O-bonded ground state and grows in with two time constants.

Page 12, paragraph 2 now reads:

“Optical studies also report a two-component rise time (72 ps and 640 ps) at wavelengths characteristic of the photoisomerization product [23]. Although a quantitative kinetic comparison of the two datasets is not particularly informative due to their different sensitivities (the optical absorption is attributed only to $^1\text{G}_\text{O}$ formation and the S K-edge difference signal is attributed to formation of all O-bonded excited and ground state species), they are consistent in reflecting the fast creation of O-bonded species. The slow rise in optical absorption was attributed to conformational relaxation of the isomerized complex; however the slow rise in S K-edge difference signal cannot be attributed to such structural changes as it is sensitive only to the Ru-S bond length which is not predicted to change significantly after formation of the $^3\text{MC}_\text{O}$ species (Supplementary Table 1, Supplementary Fig. 3).”

The authors nicely connect the charge on Ru/S with spectral shapes/shifts (p. 9: “In the MC states, the overall electron density on the Ru atom is similar to the ground state.”, p. 10: “The 1s-3p transition energy reflects the effective charge on the S atom...”). Where is the evidence for the claims of how charges (charge densities) change or do not change? The references given seem to describe very different complexes. Have local charge densities been calculated? Without quantifying these claims (albeit accepting the possible uncertainties in calculating charge densities in metal complexes) the ensuing conclusions or interpretations seem unsubstantiated.

Response 8:

The text describing changes in charge has been modified in the Ru results section to reflect changes in charge density, as opposed to discreet oxidation state changes. References have been also been added to the discussion of the Ru L-edge MLCT spectrum that explicitly address the effect of removing Ru 4d electron density upon excitation (see response 8 for text).

The changes in S charge upon Ru-S bond breaking are now quantified according to the calculated Mulliken charge.

Page 11 paragraph 3 now reads:

“For the S-bonded ground and excited state species the Ru-S bond length changes by $<3\%$ (Supplementary Table 1) and the S atomic charge density (calculated Mulliken charge) changes by $<5\%$. Therefore, it is not surprising that very small difference signals are predicted for the $^3\text{MLCT}_\text{S}$ and $^3\text{MC}_\text{S}$ states relative to $^1\text{G}_\text{S}$ (Fig. 3b). The O-bonded excited and ground states are characterized by Ru-S bond cleavage ($>45\%$ increases in Ru-S bond lengths, Supplementary Table 1) and an increased electron density on S that was previously donated to the Ru-S bond (Mulliken charge decreases 15%).”

The exemplary and important sentence that illustrates the need to validate claims made with respect to the interpretation of the x-ray absorption spectra by providing more information on the actual charge density on the respective atoms and its relation to the energies of the final (core-excited) states is on p. 8/9:

“The predicted 3MLCTS transient spectrum includes an overall spectral shift towards higher energy ... reflecting the Ru 2p core level shift that results from removal of a Ru 4d electron.” How can the authors be certain that the full equivalent of one Ru 4d electron is removed from the Ru atom? How can they be certain that this is the only cause of the spectral shift?

Response 9:

The characteristic 3MLCT spectra of several polypyridyl Ru complexes have been reported in the literature and interpreted according to similar TD-DFT calculations. The contributions of core and valence orbital shielding to the observed/calculated spectral shift have been described explicitly in the previous work and are consistent with the ³MLCT state predicted in the current manuscript. Thanks to the reviewer's comment the authors recognize that this was not acknowledged previously. A brief explanation of the predicted spectrum is now accompanied by the proper references.

Page 9 paragraph 2 now reads:

“MLCT excitation is characterized by the transfer of electron density from the Ru 4d (t_{2g}) orbitals to a bipyridyl ligand π^* orbital. The predicted ³MLCTS transient spectrum presented here is consistent with the measured and calculated ³MLCT spectra of other polypyridyl Ru complexes reported previously.[35,40] Briefly, the ³MLCT spectrum reflects the oxidation of Ru and the associated decreased shielding of the Ru 2p and e_g orbitals (manifested as an overall spectral shift towards higher energy, 2840 eV bleach and 2842 eV induced absorption), [35] and a low energy transition into the newly created t_{2g} vacancy (see Supplementary Fig. 1). The measured differential spectrum, characterized by a narrow bleach (2.841 keV) and a single broad induced absorption at lower energies (2837-2839 eV), is inconsistent with these predicted ³MLCT features.”

This may relate to the statement on p. 11 “the transient signal is not predicted to grow in magnitude upon any structural changes occurring after formation of the 3MCO state” which seems unsubstantiated as no evidence (spectra?) seems to be given.

Response 10:

This statement was based on the predicted spectra shown in Figure 3b, but we acknowledge that this would have been difficult to interpret given the misreporting of arbitrary intensity units in Figure 3. A new figure has been added to the SI (Supplemental Figure 3) that overlays the ³MC₀ and ¹G₀ difference spectra, demonstrating no change in shape or intensity upon formation of ¹G₀ from ³MC₀.

Page 13 paragraph 1 now reads:

“The slow rise in optical absorption was attributed to conformational relaxation of the isomerized complex; however the slow rise in S K-edge difference signal cannot be attributed to such structural changes as it is sensitive only to the Ru-S bond length which is not predicted to change significantly after formation of the ³MC₀ species (Supplementary Table 1, Supplementary Fig. 3).”

All discussions of spectral shifts are made in a one-electron orbital-based picture. A priori, this is often too simplified (see e.g. the spin-orbit interaction for Ru L-edge final states). Multiplet effects may be agreeably small for Ru L-edge spectroscopy but evidence has to be provided to support the one-electron picture a posteriori (to validate the approximations made implicitly). This could be done by linking better the main text to the Supplementary Information (Figs. S1 and S2). In this respect: What exactly is meant with the “Binding Energy” in Figure 1b? If it was the orbital energy it would have to increase from bottom to top, if it was minus the orbital energy, would the arrow not have to be inverted? What is meant with the dashed line?

Response 11:

The issue of one-electron assumptions has been explored extensively by others for simulations of Ru-based complexes, which is referenced on page 6, paragraph 1:

“This approach is particularly effective in 4d transition-metal complexes (e.g. Ru-based) where strong spin-orbit coupling dominates and multiplet effects are negligible, rendering TD-DFT calculations a reliable tool for quantitative predictions of X-ray spectra. [36]”

We also added the predicted transitions and predicted spectra, overlaid with the ground state measured spectra shown in Figures 3a-b, to reiterate the validity of this approach. Page 8 paragraph 2 now reads:

“The predicted transitions for the S-bonded ground state (1G_S) are overlaid with the ground state XAS data in the top panels of Fig. 3a-b, highlighting the validity of the TD-DFT simulation approach.”

Note that the questions related to former Figure 1b are now addressed by the new figures (see discussion above).

In order to better assess validity of the calculated differential spectra it would be good to see the comparison of the calculated and measured ground-state spectra (in the Supplementary Information, e.g.).

Response 12:

The calculated transitions have been overlaid with the ground state spectra in Figure 3 (sticks), as have the simulated spectrum (dashed line).

Supplementary Information:

Figures S1 and S2: Transitions are very hard to assess because line spectra are given with squares that seem to indicate the transitions. Sticks or lines would be much more useful indicating the transitions.

Response 13:

Drop lines have been added to the predicted transition data points in Supplemental Figures 1-2.

Reviewer #2:

The authors investigated electronic structures of Photochromic Ru-Sulfoxide Complex during its photoinduced isomerization processes where a Ru-S bond was transformed to a Ru-O bond accompanied by internal rotation of the S-O group as well as other parts of the molecule. Compared to previous optical studies that have been carried out by Rack's group who also synthesized the compounds, the new development is the use of TR-XAS measurements at Ru L-edge and S K-edge which are sensitive to the electronic structural changes complementary to what can be obtained from optical transient absorption measurements. They also carried out theoretical calculations to establish the correlations between the observed spectral changes with the electronic structural changes of this complex. From this study, they extracted a more detailed reaction pathways with different states which otherwise will not be extracted from previous studies.

The study has been carried out carefully, but some issues need to be addressed before it can be reconsidered as listed below.

1. The photochemical theme is unclear until the discussion part, which makes the results hard to follow with their connections to the reaction mechanisms. It at least should draw out the initially hypothesized reaction mechanisms and all the states involved at the beginning. Perhaps a reaction scheme with all the possible states involved and their presumed energy levels will help. Of course, it may run into the conflict in the discussion section where the authors currently displaced such a diagram to summarize the results.

Response:

The introduction has been re-written to include a discussion of isomerization mechanisms proposed for this class of Ru sulfoxide systems. This includes a discussion of the proposed roles of charge transfer and metal-centered states, as well as the potential energy surfaces on which isomerization occurs. See paragraphs 3-5 of introduction.

All of these adiabatic and non-adiabatic processes are also now illustrated in Figure 1b-d.

2. The authors brought up the adiabatic and non-adiabatic reaction mechanism for the isomerization process, which is very interesting to consider. However, describing such mechanisms normally requires some excited state potential energy surfaces. It is unclear what the adiabatic and non-adiabatic reaction mechanisms are defined here, and which potential energy parabola are involved. Such depiction could help to clarify the confusion. How should readers understand the non-adiabatic process that is correlated with the isomerization. It seems that the authors should clarify two sets of parameters, the pathway along the isomerization involving the nuclear motions and the energetic pathway. If they can do that the significant of the manuscript will be much higher.

Response:

The discussion of possible adiabatic and non-adiabatic isomerization pathways for this class of systems added to the introduction (see point 1 above) is also intended to address this point, as is the addition of the PES diagrams shown in Figure 1b-d.

To address the reaction mechanism of the complex measured in this work, the isomerization mechanism along the potential energy landscape is discussed in terms of the transitions and barriers between the 3MLCT and 3MC surfaces (triplet PES) in the results section and the PES calculation results are shown in Supplementary Figure 4.

Page 13 paragraph 2 reads:

“The TD-DFT calculations and analysis of the potential energy landscapes reveal low transition barriers between the 3MLCTS surface and the 3MCS (1.1 kcal/mol) and 3MCO (1.6 kcal/mol) surfaces (Supplementary Fig. 3). The possibility to populate either 3MC state upon MLCT excitation explains our observation of bimodal isomerization (two-component time response in Fig. 3d) and is consistent with the predictions of transition state theory.”

The discussion section has also been revised to expand the comparison of the isomerization mechanism of the complex measured here to other Ru sulfoxide complexes and to include a reference to the PES calculation results. See the entire last paragraph of the Discussion section.

3. The X-ray excitation diagram in Figure 1 seems unnecessary because that belongs to the definitions of L- and K-edge transitions shown in the textbook.

Response:

This figure has been replaced with Figure 2 in response to reviewer 1 (Response 2) and to clarify that the XAS measurements probes unoccupied molecular orbitals of mixed metal and ligand character (see point #4 below) , but with significant element selectivity based on the origin of the X-ray transition.

4. I would not agree with their statement: “L-edge spectroscopy probes transition-metal d orbitals, while K-edge spectroscopy probes ligand atom p orbitals”. It is a simple-minded description and could cause misleading in the community. The authors should add L-edge Mainly probes...” to be more accurate because the pre-edge in K-edge spectra could also provide information for d as observed in many studies.

Response:

This statement has been revised to indicate that any unoccupied orbital containing Ru *d* or S *p* character are probed by XAS at the Ru L-edge and S K-edge, respectively.

Page 6 paragraph 1 now reads:

“In particular, transition metal L-edge spectroscopy probes unoccupied molecular orbitals with mainly metal *d* character, while ligand K-edge spectroscopy probes molecular orbitals of predominately ligand atom *p* character (Fig. 2)...”

Also related to this point, the following descriptions of the Ru and S edge static absorption spectra have been added to page 8 paragraph 1:

“The Ru L-edge spectrum is comprised of transitions from 2*p* core levels to unoccupied orbitals of primarily Ru 4*d* character (dx^2-y^2 and dz^2), as illustrated in Supplementary Fig. 1. The S K-edge spectrum is comprised of transitions from the S 1*s* core level to antibonding orbitals of primarily sulfoxide ligand character (with some metal character contributing to the lower energy peak), as illustrated in Supplementary Fig. 2.”

5. It seems that an optical absorption spectrum in the UV/vis region will be good for readers to appreciate the work better.

Response:

A reference to the optical absorption spectrum has been added to the methods section.

Page 21 paragraph 2 now reads:

“The pump wavelength was selected in order to be close to the peak of the absorption feature attributed to MLCT excitation (see optical absorption spectrum in reference [23]).”

6. Because of the lack for introducing different states, it is hard to read time-evolution difference spectra in Figure 2. Why there are difference spectral signals for the ground state G, or if G stands for the ground state? Definitions are important.

Response:

Each possible excited state and the ground states are now defined in the Results section, page 8 paragraph 2 reads:

“The predicted differential XAS spectra for several possible excited state intermediates and the O-bonded ground state (as suggested by previous optical and DFT studies) are shown in the lower

panels of Fig. 3a-b: S-bonded MLCT state ($^3\text{MLCT}_\text{S}$), S-bonded MC state ($^3\text{MC}_\text{S}$), O-bonded MC state ($^3\text{MC}_\text{O}$) and, the O-bonded ground state ($^1\text{G}_\text{O}$) photoproduct."

However, as the reviewer suggests in point 1, the states should have been defined earlier upon discussion of various possible isomerization mechanisms. They are now defined in the Introduction section as described in point 1 above. They are also redefined in the captions of Figures 1 and 3.

We refer to two distinct ground states: the initial S-bonded ground state before isomerization ($^1\text{G}_\text{S}$) and the final O-bonded ground state after isomerization ($^1\text{G}_\text{O}$). One should expect a difference signal to occur for the final O-bonded ground state because of the major change in bonding upon isomerization. Additional definitions that will help clarify the difference in S- and O-bonded ground states have been added to the introduction and Figure 1 and 3 captions.

From the Ru perspective, only a small spectral change may be expected as its electronic configuration is unchanged between the initial and final ground states, but the Ru spectrum may be slightly shifted due to a small change in charge density in the vicinity of the central Ru. From the S perspective, a large difference signal is expected due to the major change in bonding of the S atom. This is now addressed more explicitly in page 12 paragraph 1:

"The O-bonded excited and ground states are characterized by Ru-S bond cleavage (>45% increases in Ru-S bond lengths, Supplementary Table 1) and an increased electron density on S that was previously donated to the Ru-S bond (Mulliken charge decreases 15%). These changes in bonding and charge result in large transient difference signals predicted for the $^3\text{MC}_\text{O}$ and $^1\text{G}_\text{O}$ states (Fig. 3b), with an overall spectral shift to lower energy (2476 eV bleach and 2474.5 eV induced absorption), and transient difference signal amplitudes that are more than eight times larger than those predicted for the S-bonded species ($^3\text{MLCT}_\text{S}$ or $^3\text{MC}_\text{S}$)."

7. P. 9 what is the "intermediate MC state"?

Response:

The expression above was intended to refer to the excited state reaction intermediate that is assigned as an MC state. It now instead reads "a ^3MC state is populated".

REVIEWERS' COMMENTS:

Reviewer #1 (Remarks to the Author):

The authors were very successful in my view in removing doubts and criticism related to their originally submitted manuscript. They have adequately addressed the question about how this paper would influence thinking in the field. This was achieved by a new introduction and a corresponding discussion section where potential energy curves, reaction pathways, spin states, valence charges and molecular structures are now nicely connected in a consistent fashion. This enables an accessible molecular-orbital based explanation of the observations and not only enhances impact of the study compared to the previous version of the manuscript but indeed builds the basis for sparking new thinking in the field. The x-ray spectra are now better introduced and consistently discussed. With the added MO diagram and the changes related to additional issues with data presentation and discussion all concerns were adequately met.

I can now clearly recommend publication of this version of the manuscript in Nature Communications.

Reviewer #2 (Remarks to the Author):

The authors have addressed the issues raised in the first round of the reviews sufficiently and I recommend the publication of the revised manuscript.